# Tumor Solid Stress: Assessment with MR Elastography under Compression of Patient-Derived Hepatocellular Carcinomas and Cholangiocarcinomas Xenografted in Mice

**DOI:** 10.3390/cancers13081891

**Published:** 2021-04-15

**Authors:** Gwenaël Pagé, Marion Tardieu, Jean-Luc Gennisson, Laurent Besret, Philippe Garteiser, Bernard E. Van Beers

**Affiliations:** 1Laboratory of Imaging Biomarkers, Center of Research on Inflammation, Université de Paris, UMR 1149, Inserm, F-75018 Paris, France; philippe.garteiser@inserm.fr (P.G.); bernard.van-beers@aphp.fr (B.E.V.B.); 2Montpellier Cancer Research Institute (IRCM), INSERM U1194, University of Montpellier, 34095 Montpellier, France; marion.tardieu@umontpellier.fr; 3Montpellier Cancer Institute (ICM), 34298 Montpellier, France; 4Université Paris-Saclay, CEA, CNRS, Inserm, BioMaps, 91401 Orsay, France; jean-luc.gennisson@universite-paris-saclay.fr; 5Sanofi R&D, F-94400 Vitry-sur-Seine, France; laurent.besret@sanofi.com; 6Department of Radiology, AP-HP, Beaujon University Hospital Paris Nord, F-92110 Clichy, France

**Keywords:** tumor pressure, interstitial fluid pressure, tumor viscoelasticity, solid stress, magnetic resonance elastography

## Abstract

**Simple Summary:**

Tumor biomechanical properties, including high viscoelasticity and tumor pressure (solid stress and interstitial fluid pressure), are linked to tumor severity. While tumor viscoelasticity can be quantified with MR elastography, a non-invasive method to quantify tumor pressure remains elusive. In patient-derived hepatocellular carcinomas and cholangiocarcinomas xenografted in mice, we observed that basal elasticity determined during MR elastography under compression had high diagnostic performance in assessing tumor fibrosis content and was independently influenced by interstitial fluid pressure. In contrast, compression stiffening rate had high diagnostic performance in assessing solid stress. Assessment of compression stiffening with MR elastography may provide a non-invasive biomarker of tumor solid stress.

**Abstract:**

Malignant tumors have abnormal biomechanical characteristics, including high viscoelasticity, solid stress, and interstitial fluid pressure. Magnetic resonance (MR) elastography is increasingly used to non-invasively assess tissue viscoelasticity. However, solid stress and interstitial fluid pressure measurements are performed with invasive methods. We studied the feasibility and potential role of MR elastography at basal state and under controlled compression in assessing altered biomechanical features of malignant liver tumors. MR elastography was performed in mice with patient-derived, subcutaneously xenografted hepatocellular carcinomas or cholangiocarcinomas to measure the basal viscoelasticity and the compression stiffening rate, which corresponds to the slope of elasticity versus applied compression. MR elastography measurements were correlated with invasive pressure measurements and digital histological readings. Significant differences in MR elastography parameters, pressure, and histological measurements were observed between tumor models. In multivariate analysis, collagen content and interstitial fluid pressure were determinants of basal viscoelasticity, whereas solid stress, in addition to collagen content, cellularity, and tumor type, was an independent determinant of compression stiffening rate. Compression stiffening rate had high AUC (0.87 ± 0.08) for determining elevated solid stress, whereas basal elasticity had high AUC for tumor collagen content (AUC: 0.86 ± 0.08). Our results suggest that MR elastography compression stiffening rate, in contrast to basal viscoelasticity, is a potential marker of solid stress in malignant liver tumors.

## 1. Introduction

It is increasingly recognized that biomechanics of solid tumors affects their growth, invasion, and resistance to treatment [1,2,3,4]. The adverse role of tumor biomechanical properties has been observed in various tumors, including carcinomas of the liver, pancreas, breast, and colon [1,5,6,7,8]. The abnormal biomechanical characteristics of malignant tumors include high stiffness or viscoelasticity, solid stress, and interstitial fluid pressure.

High viscoelasticity is mainly caused by collagen deposition and cross-linking but also by high interstitial fluid pressure [9,10,11,12]. In contrast, high solid stress is related to cell proliferation, matrix deposition, proteoglycan gel swelling, and resistance to the volume expansion from outside the tumor. Elevated interstitial fluid pressure is explained by the lack of functional lymphatic vessels and the abnormal capillary leakiness in tumors [13]. The altered tumor biomechanical properties promote tumor progression and treatment resistance [13,14].

The biomechanical properties are classically measured ex vivo in tissue samples. Viscoelasticity is quantified with rheological measurements, whereas cutting methods are used to assess solid stress [15,16]. In vivo, insertion of wick-in needles or more recently piezoelectric pressure catheters is performed to measure the tumor pressure components, including the solid stress and interstitial fluid pressure [17].

MR elastography is increasingly used to non-invasively assess the tissue stiffness (i.e., the shear modulus G*) or viscoelasticity (i.e., the storage modulus G’ corresponding to the elasticity and the loss moduli G”, corresponding to the viscosity) in vivo [9,11,18,19]. However, the role of elastography in assessing tumor solid stress remains debated. Indeed, even if high collagen is a common cause for high stiffness and solid stress, tumors with high solid stress are not necessarily stiff [2].

Recently, investigators have used elastography to assess the behavior of biomechanical properties under static load. It has been observed that the apparent stiffness increases under compression. This phenomenon, known as compression stiffening, has been observed in liver tissue and tumors [20,21,22,23,24,25]. Moreover, it has been suggested that the evolution of the stiffness under compression (the compression stiffening rate) is related to solid stress [13,26,27]. However, this hypothesis has not been validated with in vivo MR elastography, to the best of our knowledge. Therefore, the aim of our study was to study the potential role of MR elastography at the basal state and under compression to assess the biomechanical properties of hepatic tumors, including their solid stress in living mice.

## 2. Results

### 2.1. The Repeatability Index of the MR Elastography Measurements in Malignant Hepatic Tumors Is about 20%

To assess the biomechanical properties of malignant liver tumors with MR elastography, we used two malignant liver tumor models, namely patient-derived hepatocellular carcinomas and cholangiocarcinomas xenografted subcutaneously in the right flank of severe combined immunodeficient mice. Suitable measurements of the compression stiffening rate with MR elastography could be obtained in 15 of the 19 mice with hepatocellular carcinomas and in 9 of the 10 mice with cholangiocarcinomas. These 24 mice constituted the final study group. In the five remaining mice, tumor displacement during compression prevented correct assessment of the evolution of mechanical properties. The repeatability indexes of the elasticity, the viscosity, and the compression stiffening rate measurements were 22%, 23%, and 24%, respectively.

### 2.2. The Hepatocellular Carcinomas and Cholangiocarcinomas Have Different MR Elastography Pressure and Histological Characteristics

We compared the MR parameters, pressure measurements, and histological features between hepatocellular carcinomas and cholangiocarcinomas. According to MR elastography, the elasticity and viscosity were significantly lower in the hepatocellular carcinomas than in the cholangiocarcinomas (1.6 ± 0.3 kPa versus 2.2 ± 0.5 kPa and 1.1 ± 0.2 kPa versus 1.8 ± 0.4 kPa, *p* < 0.001). The compression stiffening rate (slope of elasticity versus applied compression, Figure 1) did not differ significantly between the hepatocellular carcinomas and the cholangiocarcinomas (0.6 ± 0.6 adim versus 0.8 ± 0.8 adim, *p* = 0.49). The solid stress, measured with the piezoelectric pressure catheter, was significantly lower in the hepatocellular carcinomas than in the cholangiocarcinomas (10.7 ± 4.3 mmHg versus 15.8 ± 6.6 mmHg, *p* = 0.04). In contrast, the interstitial fluid pressure did not differ significantly between the two tumor types (2.3 ± 0.9 mmHg versus 2.3 ± 1.0 mmHg, *p* = 0.77). The size of the tumor nodules was similar in the hepatocellular carcinomas (726 ± 210 mm^3^) and the cholangiocarcinomas (809 ± 222 mm^3^, *p* = 0.60) (Figure 1).

According to digital histopathology tests, the tumors did not contain necrotic areas. The hepatocellular carcinomas had significantly lower collagen content and higher cellularity than the cholangiocarcinomas did (1.1 ± 0.4% versus 3.6 ± 0.8%, *p* < 0.001 and 252 ± 10 cells/mm^2^ versus 233 ± 16 cells/mm^2^, *p* = 0.005, respectively) (Figure 2).

### 2.3. Interstitial Fluid Pressure Is an Independent Determinant of Basal Viscoelasticity, whereas Solid Stress Is an Independent Determinant of Compression Stiffening Rate

We looked at the associations between the MR elastography parameters and tumor pressure, volume, and histology with partial correlation and multiple regression analysis. When taking tumor type as covariate, the interstitial fluid pressure and the collagen content were both significantly correlated with the elasticity and viscosity (*r* = 0.70, *p* < 0.001; *r* = 0.84, *p* < 0.001 for elasticity, and *r* = 0.76, *p* < 0.001; *r* = 0.75, *p* < 0.001 for viscosity, respectively). In contrast, no significant correlation was observed between the solid stress, the tumor volume, and the cellularity on one hand, and the viscoelasticity on the other (Table 1).

The solid stress, the tumor volume, the collagen content, and the cellularity were correlated with the compression stiffening rate (*r* = −0.67, *p* < 0.001; *r* = −0.41, *p* = 0.04; *r* = −0.55, *p* = 0.007; and *r* = 0.59, *p* = 0.003, respectively). In contrast, the interstitial fluid pressure was not significantly correlated to the compression stiffening rate (Table 1).

The results of multiple regression showed that interstitial fluid pressure and mainly collagen content were determinants of elasticity (*r*_partial_ = 0.63, *p* = 0.001; *r*_partial_ = 0.86, *p* < 0.001, respectively) and viscosity (*r*_partial_ = 0.62, *p* = 0.002; *r*_partial_ = 0.91, *p* < 0.001, respectively). Solid stress (*r*_partial_ = −0.67, *p* < 0.001), in addition to collagen content (*r*_partial_ = −0.54, *p* = 0.01), cellularity (*r*_partial_ = −0.58, *p* = 0.006), and tumor type (*r*_partial_ = −0.76, *p* = 0.0001), was an independent determinant of the compression stiffening rate (Table 1).

### 2.4. In MR Elastography the Compression Stiffening Rate Has High Performance in Diagnosing Elevated Solid Stress

We next assessed the diagnostic accuracy of the MR elastography parameters for diagnosing high solid stress (>15.6 mm Hg) and substantial collagen content. The compression stiffening rate had high performance in diagnosing elevated solid stress (area under the receiver operating characteristic curve (AUC): 0.86 ± 0.08, *p* < 0.001). In contrast, the AUC of elasticity for diagnosing elevated solid stress was not statistically significant (AUC: 0.63 ± 0.12, *p* = 0.30) (Figure 3).

For tumor collagen content, the elasticity had high diagnostic performance (AUC: 0.86 ± 0.08, *p* < 0.001), in contrast to the compression stiffening rate, for which AUC was not significant (AUC: 0.59 ± 0.12, *p* = 0.48).

## 3. Discussion

Our study highlights the relationships between the mechanical parameters determined by MR elastography and the physical microenvironment in hepatic tumors. In multiple regression, we observed that the tumor collagen content and, to a lesser degree, the interstitial fluid pressure were determinants of basal viscoelasticity according to MR elastography, whereas the tumor solid stress, in addition to the collagen content, cellularity, and tumor type, was an independent determinant of the compression stiffening rate. The compression stiffening rate had high diagnostic performance in diagnosing elevated solid stress, whereas basal elasticity had high diagnostic performance in diagnosing tumor collagen content.

Our results for basal MR elastography are in accordance with previously reported results. Indeed, viscoelasticity is an established marker of collagen content in chronic liver disease and tumors [9,12]. Moreover, it is known that, apart from this static component of liver stiffness (i.e., collagen fraction and crosslinking), changing fluid pressure related to various conditions, including hepatic necro-inflammation, edema, congestion, eating, arterial, and portal venous hypertension, may modulate liver stiffness [28,29].

In contrast to this known link between stiffness on one hand and fibrosis and interstitial fluid pressure on the other, the correlation between viscoelasticity and solid stress remains debated. Tumor stiffness, assessed with ultrasound elastography, has been proposed as a surrogate marker of solid stress [6]. However, in an ex vivo study, Nia H.T. et al. showed that tumors can have substantially different levels of solid stress despite similar stiffness values [2]. These results agree with our in vivo findings where the solid stress was not correlated with the basal viscoelasticity but with the compression stiffening rate.

Compression stiffening has been repeatedly described in phantoms and biological tissues [20,21,22,23,24,26]. Compression stiffening may create diagnostic problems when the added stress is unknown because it can result in loading bias, an apparent stiffness increase that is a conglomerate measure of the intrinsic material properties and the effect of external compression [24]. In contrast, it has been reported that changes in apparent tumor stiffness under defined compression may improve tumor detection and characterization [30].

Moreover, it has been suggested that this compression stiffening mechanism is a way to assess solid stress with elastography [13,26,27], as observed in our study. Interestingly, it has been shown that a combination of nonlinear polymer network and particle inclusions is essential to mimic compression stiffening that cannot be reproduced by either biopolymer networks or colloidal particle systems alone [16]. The combined effect of biopolymer network and particles on compression stiffening parallels our findings as we observed that both the tumor collagen content and cellularity were independent determinants of the measured compression stiffening rate.

Our findings are also in agreement with those of Nia H.T. et al., who showed in their ex vivo study that solid stress depends on both cancer cells and microenvironment [2]. In that study, it was also observed that solid stress increased with tumor size. Similarly, in our study, we observed a significant correlation between compression stiffening rate and tumor volume. However, tumor volume was not an independent determinant of the compression stiffening rate in multiple regression analysis.

Moreover, it should be noted that even if both collagen content and cellularity were determinants of the compression stiffening rate in our study, a negative correlation between collagen content and compression stiffening rate was observed as expected, but a positive correlation appeared between cellularity and compression stiffening rate. This last feature suggests that tumors with high cellularity have high compression stiffening rate, meaning that they have low solid stress. This inverse correlation between cellularity and solid stress was also shown in our study by the fact that the cholangiocarcinomas, which had higher solid stress than the hepatocellular carcinomas, also had higher collagen content but lower cellularity than the hepatocellular tumors.

The inverse correlation between cellularity and solid stress may seem counterintuitive, as solid stress originates from solid tumor components, including cells, collagen, and hyaluronic acid [31]. However, this inverse correlation may be explained by several factors. First, it has been reported that elevated solid stress can reduce cancer cell proliferation rate and induce apoptosis [31]. Second, solid stress is not only influenced by forces created by tumor cells and microenvironment but also by compressive stress from surrounding tissue. Third, the influence of tumor cells, microenvironment, and surrounding tissue on solid stress may depend on tumor type and location, as shown by previously reported differences in solid stress between primary tumors and their metastases [2]. The influence of tumor cells, microenvironment, and surrounding tissue on solid stress and compression stiffening rate should be further studied in various tumor types. The compression stiffening rate as measured with MR elastography in our study made it possible to differentiate between tumors with low and high solid stress, independently of their stiffness. Diagnosing high solid stress with MR elastography was feasible in the two types of malignant hepatic tumors that differed regarding their solid stress, cellularity, and collagen content. Moreover, we observed satisfactory repeatability measurements of the compression stiffening rate and basal viscoelasticity measurements using MR elastography with repeatability coefficients of 22–24%. Our results of high diagnostic accuracy and repeatability suggest that the compression stiffening rate may become a valid marker of high solid stress.

Our results may have prognostic significance, as high solid stress has been shown to promote tumor progression and hinder the delivery and efficacy of anti-cancer therapies by compressing blood and lymphatic vessels [32]. The translation of the elastography method to assess tumor solid stress in clinical studies is currently underway. The feasibility of measuring compression stiffening with ultrasound and MR elastography in patients with liver, brain, and breast tumors has already been reported in preliminary studies [30,33].

Our study has limitations. First, the compression stiffening rate, as assessed in our study, depends not only on the tumor forces but also on the angulation of the exerted stress [34]. This angulation could not be standardized by balloon compression in our study. Therefore, the compression stiffening rate, as assessed here, should be regarded as a semiquantitative marker of high solid stress. The changes of the mechanical properties under uniaxial stress have been reported in several studies [16,26,30]. Uniaxial compression, performed with rigid plates, simplifies the mathematical theory of the relation between solid stress and mechanical properties [26] but may be challenging to obtain, especially in deep-seated organs.

Second, correct MR elastography measurements could not be obtained in five mice because the balloon slipped away during compression. This technical problem was mainly observed in the first part of the study and was related to the experience of the examinator. Here again, the use of rigid plates for compression might alleviate the problem.

Third, the tumors in our study had low fibrosis content (<5%). Higher stroma content (>50%) has been described in stroma rich hepatocellular carcinomas and cholangiocarcinomas in patients [35,36]. Because of the low collagen content in the tumors of our study, we had to define a low internal cutoff (1.7%) for substantial fibrosis. MR elastography under compression should be further assessed in tumors with high desmoplasia, including pancreatic and breast cancer [37].

In clinical practice, one area in which compression MR elastography might be particularly useful is the assessment of tumor response to various treatments, including immunotherapy. Recently, immunotherapy based on immune checkpoint blockers has been increasingly tested in clinical trials of cancers, including advanced hepatocellular carcinomas [38]. However, the therapeutic success rate of single immune checkpoint blockers remains limited [39]. The aberrant tumor microenvironment has been identified as one cause of resistance to this immunotherapy. Moreover, targeting nonimmune components of the tumor microenvironment by normalizing or decompressing the vasculature with anti-VEGF antibodies and renin-angiotensin system inhibitors, respectively, represents a clinically translatable strategy to overcome resistance to immune checkpoint blockers [38,40,41] Compression MR elastography, as developed in our study, might prove useful to assess the normalization of the tumor biomechanical properties, including solid stress, by these treatments.

Finally, hepatocellular carcinomas mainly occur in chronic liver diseases in which there is a context of chronic inflammation, fibrosis, and cirrhosis. This fibro-inflammation changes the immune state of the liver and influences the response of hepatocellular carcinomas to immunotherapy [42]. MR elastography can be used to assess the biomechanical changes in both liver and tumor microenvironment [12,18].

## 4. Materials and Methods

### 4.1. Patient-Derived Tumors Xenografted in Mice

Two tumor models, patient-derived hepatocellular carcinoma xenografts (SA-LIV-0110) and patient-derived cholangiocarcinoma xenografts (SA-LIV-0030), obtained from the Sanofi oncology biobank (Sanofi, Vitry-sur-Seine, France), were used in the study. The tumors were implanted subcutaneously in the right flank of female, 6-week-old, severe combined immunodeficient mice (Charles River, Ecully, France). Implantation of hepatocellular carcinomas was performed in 20 mice and of cholangiocarcinomas in 10 mice. After tumor implantation, the mice were transferred from the Sanofi animal facilities to those of Bichat medical school, Université de Paris. The mice were housed in laboratory cages with aspen bedding (Tapvei Oyj, Kaavi, Finland) and cage enrichment with a maximum of five animals per cage. They were fed regular mouse diet from Altromin (Lage, Germany). When the tumors reached a volume >200 mm^3^ (4 to 8 weeks after implantation), MR imaging was performed during anesthesia with isoflurane in a 70:30 air/O_2_ mixture. MR imaging was obtained during light cycle, and mice were not fasting. Tumor growth was not observed in one mouse with hepatocellular carcinoma. Therefore, MR imaging was performed in 19 mice with hepatocellular carcinomas and 10 mice with cholangiocarcinomas.

### 4.2. Compression Setup for MR Elastography

A previously described method [22] was used to compress the tumors. The tumors were centered on a plastic piston. Mechanical vibrations were generated with an electromagnetic shaker (Exciter type 4808, Brüel & Kjaer, Nærum, Denmark) and transmitted to the tumors via a flexible carbon fiber rod linked to the piston [43]. A rubber balloon was placed at the opposite side of the tumors and maintained with a taped gauze strip. The balloon was inflated with air to exert compression on the tumors. A sensor was placed between the balloon and the tumors to measure the exerted stress. The sensor was connected to a pressure transducer (MPXV7007GP, NXP Semiconductors, Eindhoven, The Netherlands) interfaced to an Arduino system for recording purposes.

### 4.3. MR Imaging

MR imaging of the tumors was performed on a 7T MR scanner (Pharmascan, Bruker, Ettlingen, Germany) with a 72 mm inner diameter volume resonator and a 20 mm diameter circular surface coil. The imaging protocol included the following sequences. First, a rapid acquisition with relaxation enhancement spin echo sequence was performed to obtain high-resolution T_2_-weighted images of the whole tumor. The acquisition parameters included 60 ms echo time (TE), 200 × 200 × 200 µm^3^ resolution, and 130 × 100 matrix size. The number of slices, repetition time (TR), and acquisition time were determined by tumor size (typically, the slice number was 90, TR 13000 ms, and acquisition time 2 min 30 s).

MR elastography was performed with a mechanical excitation frequency of 600 Hz synchronized with a spin-echo imaging sequence. Sinusoidal motion encoding gradients with a maximum amplitude of 300 mT/m and a duration of two mechanical periods were used. MR elastography acquisitions were performed with 1007 ms TR, 87 ms TE, 87 × 67 matrix, 9 slices, 300 × 300 × 350 µm^3^ spatial resolution, and 4 min 30 s scan time. The MR elastography sequence was repeated in each of the three spatial directions. Four different offsets were recorded in successive acquisitions. MR elastography was performed at basal state and after inflating the balloon with 1.5 and 3 mL of air.

### 4.4. MR Image Analysis

Tumor volumes were manually delineated on the T2-weighted MR images. Each pixel of the MR elastography images was processed to calculate the storage modulus (G’ in kPa, termed “elasticity”) and loss modulus (G” in kPa, termed “viscosity”) by inversion of the Helmholtz wave, Equation (1)
(1)$$G^{*}=-\frac{\rho \omega^{2}\vec{q}}{\nabla^{2}\vec{q}},\;\;G^{*}=G^{\prime }+iG\prime \prime$$
carried out on the complex-valued curl ($\vec{q})$ of the displacement field ($\vec{u}$) obtained by unwrapping and filtering the individual MR phase images for each encoded direction and at each mechanical time offset, where ρ is the density of the material, ω the angular frequency, G* the complex shear modulus (with |G*| termed “stiffness”), and $\nabla^{2}$ the Laplace operator [44]. Elasticity and viscosity were displayed as parametric maps. Considering the curl operation used in the reconstruction method [45], measurements were taken from the five central slices of the MR elastography acquisitions.

The variation of tumor elasticity with exerted compression was quantified as the slope (in a-dimensional units (adim)) of the linear regression between elasticity (kPa) and compression (kPa) for each mouse. This slope was termed “compression stiffening rate”. The compression stiffening rate can be viewed as the ease with which tumor elasticity can be altered when compressed.

Average values of the mechanical parameters were obtained from regions of interest (ROIs) positioned manually on the tumors, avoiding their borders. The ROIs were drawn on the magnitude images obtained with the MR elastography acquisition and were copied on the corresponding parametric maps.

The repeatability of the viscoelasticity and compression stiffening rate measurements according to MR elastography was assessed in five mice with hepatocellular carcinomas. In these mice, the MR elastography examinations were performed twice at 24 h intervals. The repeatability index was calculated as 1.96 × standard deviation (%) [46]. The MR image analysis was performed by two researchers with PhDs (M.T. and G.P) and 8 and 7 years of experience in MR imaging, and both were blinded to the results of the histological analysis.

### 4.5. Tumor Pressure Measurements

After MR imaging, the mice were placed outside the MR scanner and kept under anesthesia. The total tumor pressure was measured by inserting a catheter-mounted piezoelectric pressure transducer (SPR-1000 Mikro-Tip, Millar Instruments, Houston, TX, USA) in the center of the tumors (Figure 4). Afterwards, the interstitial fluid pressure was measured by covering the pressure catheter with a perforated 24-gauge polytetrafluoroethylene sheath (Cole-Palmer, Vernon Hills, IL, USA) as described by Nieskoski M.D. et al. [17]. Solid stress was calculated by subtracting the interstitial fluid pressure from the total pressure. The pressure components were measured twice and averaged.

### 4.6. Digital Histological Image Analysis

After the pressure measurements, the mice were sacrificed, and the tumors were excised and fixed in formalin solution (Figure 5). To compare the histological analysis with the MR imaging parameters, the tumor samples were aligned relative to their position inside the MR imaging system [9] and embedded in paraffin. Tissue sections of 5 µm were cut in the same orientation as the MR imaging slices and used for histologic analysis.

Fibrosis and necrosis were assessed on hematoxylin/eosin stained sections. Collagen I and III were further quantified after picrosirius red staining. The whole tissue sections were digitized with an Aperio Scanscope (Aperio Technologies, Vista, CA, USA) at ×20 magnification. A homemade semi-automated algorithm developed with the Image J software (version 1.51, provided in the public domain by the National Institutes of Health, Bethesda, MD, USA; http://www.imagej.nih.gov, accessed on 20 March 2021, RRID:SCR_003070) was used to quantify the collagen fraction of the complete tissue section. Red, green, blue deconvolution was performed on the image, and a threshold was applied on the red image to segment the picrosirius stained areas.

Cellularity was quantified after 4′, 6-diamidino-2-phenylindole (DAPI) staining (ProLong Gold Antifade reagent with DAPI, Thermo Fisher Scientific, Bremen, Germany). Using an electronic camera (Axiocam, Zeiss, Jena, Germany), snapshots at ×200 magnification were taken across the tumor. The nuclei were counted using a plugin written with the Image J software. The cell nuclei were extracted using the Otsu thresholding algorithm, then a watershed algorithm was applied to distinguish each nucleus [47].

### 4.7. Statistical Analysis

The sample size was calculated to allow at least 80% power to detect at a 5% significance level, a significant difference between the AUC of the compression stiffening rate, and a null hypothesis value of 0.50, considering the AUC of the compression stiffening rate being 0.85 for high solid stress (≥15.6 mm Hg), and the ratio of mice with low to high solid stress being 1.7. This ratio was obtained from preliminary pressure measurements performed in eight mice (five with hepatocellular carcinomas and three with cholangiocarcinomas). For the power calculation, mice with hepatocellular carcinomas and cholangiocarcinomas were considered together as we aimed to validate the use of the compression stiffening rate as a marker of solid stress in different tumors. Under these conditions, at least 22 mice were required in the study. A total number of 30 mice was used to compensate for possible 1/3 loss of mice.

Statistical differences of tumor features between the 15 mice with hepatocellular carcinomas and 9 mice with cholangiocarcinomas were determined with the Mann–Whitney U test. The associations between the tumor pressure, volume, and histology on one hand and the mechanical properties as determined by MR elastography on the other hand were assessed with partial correlations, taking tumor type (hepatocellular carcinoma versus cholangiocarcinoma) as covariate, and with multiple regression using a stepwise method.

The performance of elasticity and compression stiffening rate to diagnose high tumor solid stress and collagen content was assessed by calculating their AUCs. A solid stress threshold of 15.6 mmHg, corresponding to the reported pressure at which vessels collapse [48,49], was used to define high solid stress. A cutoff of 1.7%, corresponding to the value that best discriminated between hepatocellular carcinoma and cholangiocarcinoma collagen content in our study, was used to define substantial collagen content. Statistical analysis was performed with the Medcalc version 19.5.3 software (Medcalc, Ostend, Belgium, RRID:SCR_015044), and statistical significance was set at *p* < 0.05.

## 5. Conclusions

In conclusion, the results of our study suggest that the compression stiffening rate as determined by MR elastography is a potential marker of solid stress in malignant liver tumors. Compression MR elastography should be further assessed as marker of prognosis and response to treatment in malignant tumors.

## Figures and Tables

**Figure 1 cancers-13-01891-f001:**
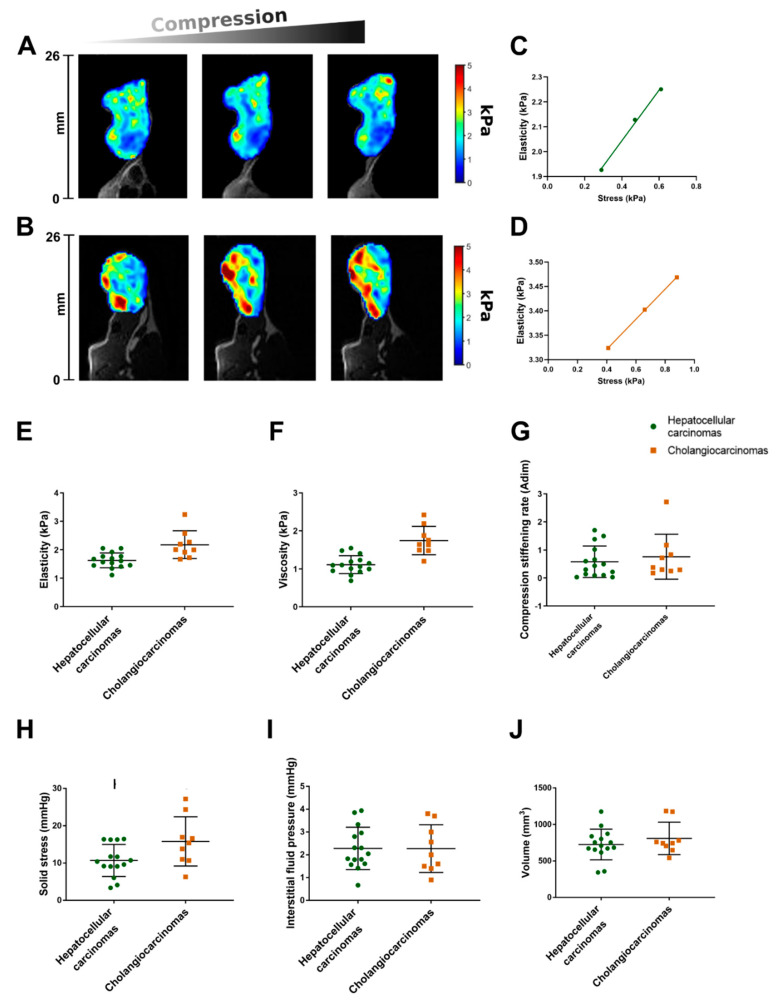
Mechanical properties (**A**–**F**), pressure measurements (**G**,**H**), and tumor volume (**I**) in hepatocellular carcinomas and cholangiocarcinomas. Elasticity color maps of hepatocellular carcinoma (**A**) and cholangiocarcinoma (**B**) from basal state (left) to maximum compression (right). Apparent elasticity increases with compression. (**C**) Corresponding compression stiffening rates in the hepatocellular carcinoma ((**C**), 1 adim) and the cholangiocarcinoma ((**D**), 0.7 adim). Box and scatter plots of mechanical properties show that elasticity and viscosity are significantly lower in hepatocellular carcinomas than in cholangiocarcinomas (*p* < 0.001) (**E**,**F**). Compression stiffening rate does not significantly differ between hepatocellular carcinomas and cholangiocarcinomas (*p* = 0.49) (**G**). Solid stress is significantly lower in hepatocellular carcinomas than in cholangiocarcinomas (*p* = 0.04) (**H**). Interstitial fluid pressure and tumor volume do not significantly differ between hepatocellular carcinomas and cholangiocarcinomas (*p* = 0.77 and *p* = 0.60, respectively) (**I**,**J**). Middle horizontal bar on graphs represents median. Lower and upper horizontal bars represent 25th and 75th percentiles.

**Figure 2 cancers-13-01891-f002:**
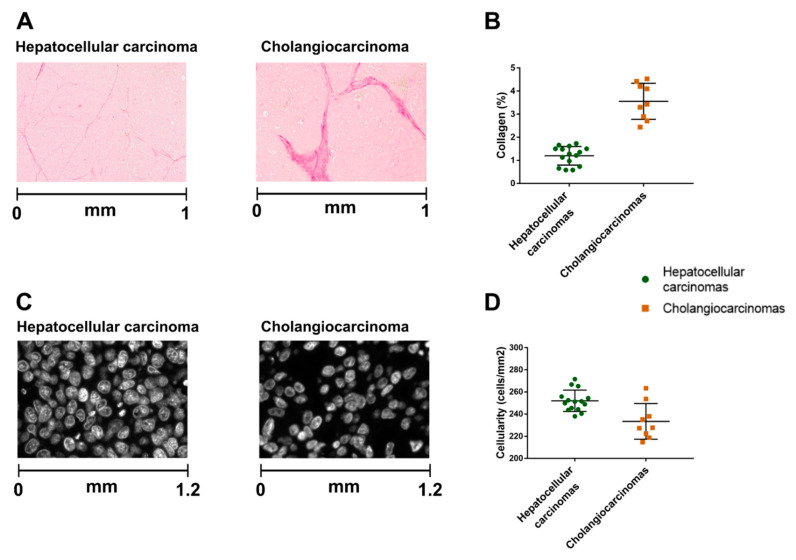
Comparison of digital histological features between hepatocellular carcinomas and cholangiocarcinomas. Representative picrosirius red pictures to quantify collagen (**A**) and nucleus stained pictures (**C**) to quantify cellularity. Hepatocellular carcinomas have significantly lower collagen content and higher cellularity than cholangiocarcinomas (*p* < 0.001 and *p* = 0.005, respectively) (**B**,**D**).

**Figure 3 cancers-13-01891-f003:**
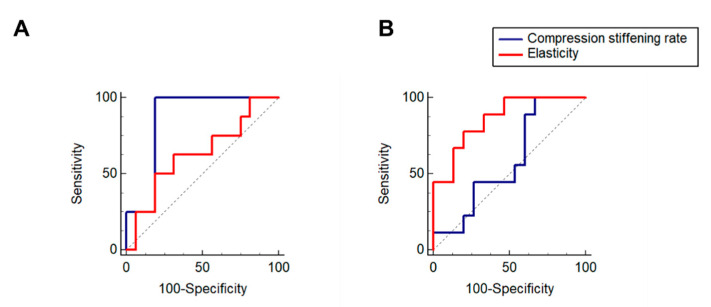
(**A**). Receiver operating characteristic curves of compression stiffening rate and elasticity for high solid stress (**A**) and substantial collagen content (**B**). Compression stiffening rate has high performance in diagnosing elevated solid stress (AUC: 0.86 ± 0.08), in contrast to elasticity (AUC: 0.63 ± 0.12) (**A**). Elasticity has high performance in diagnosing collagen content (AUC: 0.86 ± 0.08), in contrast to compression stiffening rate (AUC: 0.59 ± 0.12) (**B**).

**Figure 4 cancers-13-01891-f004:**
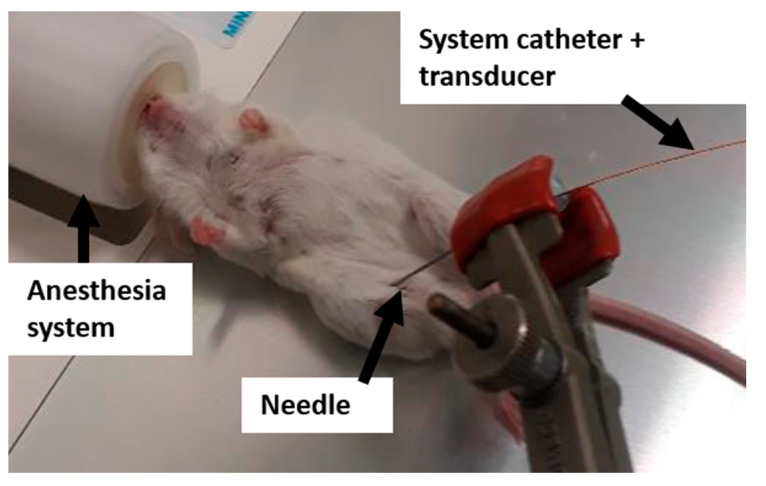
Experimental setup to measure tumor pressure with a catheter-mounted piezoelectric transducer.

**Figure 5 cancers-13-01891-f005:**
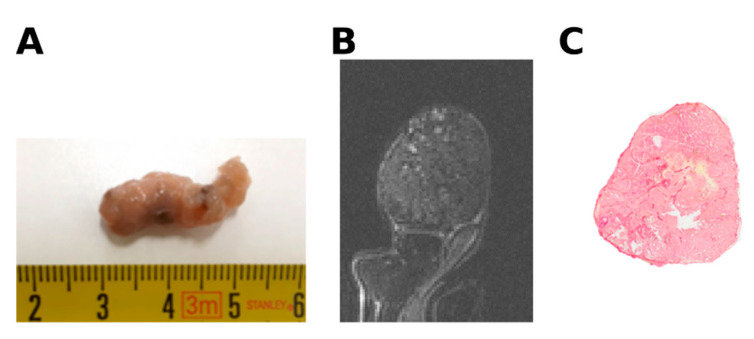
(**A**) Macroscopic view of excised tumor (**B**); tumor T2-weighted image. (**C**); corresponding picrosirius red stained section.

**Table 1 cancers-13-01891-t001:** Associations between mechanical parameters determined by MR elastography and tumor pressure, volume, and histological features.

Biomechanical Parameter	Partial Correlation with Tumor Type as Covariate		Multiple Regression			
	*r*	*p*	β Value	*r* _partial_	*p*	R^2^
**Basal elasticity** **(*n* = 24)**						0.82
Solid stress	−0.07	0.74	NI			
Interstitial fluid pressure	0.70	<0.001	0.17	0.63	0.001	
Volume	0.20	0.35	NI			
Collagen	0.84	<0.001	0.28	0.86	<0.001	
Cellularity	−0.27	0.22	NI			
Tumor type	-	-	NI			
**Basal viscosity** **(*n* = 24)**						0.87
Solid stress	−0.02	0.91	NI			
Interstitial fluid pressure	0.76	<0.001	0.13	0.62	0.002	
Volume	−0.19	0.38	NI			
Collagen	0.75	<0.001	0.30	0.91	<0.001	
Cellularity	−0.26	0.24	NI			
Tumor type	-	-	NI			
**Compression stiffening rate (*n* = 24)**						0.75
Solid stress	−0.67	<0.001	−0.06	−0.67	<0.001	
Interstitial fluid pressure	−0.07	0.75	NI			
Volume	−0.41	0.04	NI			
Collagen	−0.55	0.007	−0.44	−0.54	0.01	
Cellularity	0.59	0.003	0.02	0.58	0.006	
Tumor type	-	-	−1.80	0.76	<0.001	

NI: Not included in model. Bold: To separate each parameter analyzed.

## Data Availability

The data presented in this study are openly available on request from the corresponding author.

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
