# Peer review of "Tumor Solid Stress: Assessment with MR Elastography under Compression of Patient-Derived Hepatocellular Carcinomas and Cholangiocarcinomas Xenografted in Mice"

_cancers, 2021, doi:10.3390/cancers13081891_

Round 1

Reviewer 1 Report

In this manuscript, Gwenaël Pagé et al. performed MR elastography was performed in mice with a patient-derived, subcutaneously xenografted HCC and cholangiocarcinoma model. Although the manuscript has provided interesting results, it still has room for improvement.

Major points:

  1. The authors employed the Mann-Whitney U test. Some advantages of this tool are the chance of detecting difference as significant or occurred by chance, showing the median between 2 sets of data, using data sets of different sizes. Mann-Whitney U test is good with dealing with skewed data, so data doesn't need to be normally distributed. Nonetheless, the disadvantages are that lengthy calculation - prone to human error. Does not explain why there is a difference. Moreover, the Mann-Whitney U test is more appropriate when the data sets are independent of each other. More appropriate when both sets of data have the same shape distribution. Additionally, become less accurate when the sample size is below 5 or above 20 use only the mentioned dataset. Can the authors comment on this topic?
  2. AUC ROC is essentially the measurement of the order of the predictions. There is a way to trick the AUC ROC metric. If we divide all of the predictions by 100, we will keep the same AUC ROC. Despite the fact that AUC ROC is a really reliable metric, we do care about the actual probability we are giving, instead of just the ranking. Brier Score is a great supplement to AUC ROC. Could the author implement, or, conversely, explain?
  3. How would the author translate their findings into every day clinical practice?
  4. In the frame of this thinking, the authors corrected cited the role of tumour microenvironment in mediating the solid stress (reff. 2, 8). I personally miss some important references regarding cancer niche in the liver and their therapeutic consequences besides the physical component. Indeed, Tregs and Th17 cells are both found to be increased in more advanced HBV-related fibrosis compared to earlier stage fibrotic livers; however, an elevated Th17/Treg ratio has been shown to correspond with higher liver stiffness measurement, a correlate of worsening liver fibrosis (PMID: 31627433). Thus, since nivolumab and pembrolizumab, two therapeutics against the Programmed death (PD)-ligand 1 (PD-L1)/PD1 axis have been recently approved for subsequent-line therapy. However, similar to other solid tumours, the response rate of single-agent targeting PD-L1/PD1 axis is low Therefore, a lot of combinatory approaches are under investigation, including the combination of different immune checkpoint inhibitors (ICIs), the addition of ICIs after resection or during loco-regional therapy, ICIs in addition to kinase inhibitors, anti-angiogenic therapeutics, and others (PMID: 31627433). In light of the authors' findings, these insights might prompt further therapeutic scenarios. The authors should expand on this topic, considering the issues mentioned.

Reviewer 2 Report

In this manuscript entitled “Tumor solid stress: assessment with MR elastography under compression of patient-derived hepatocellular carcinomas and cholangiocarcinomas xenografted in mice”, Pagé et al demonstrated MR elastography compression stiffening rate, in contrast to basal viscoelasticity, is a potential marker of solid stress in malignant liver tumors. Overall, experiments were well designed, performed and data generated in this manuscript partially supports study. 

Here are comments:

  1. The authors observed that “Solid stress measured with the piezoelectric pressure catheter was significantly lower in the hepatocellular carcinomas than in the cholangiocarcinomas and hepatocellular carcinomas had significantly lower collagen content and higher cellularity than the cholangiocarcinomas did”. Authors should discuss these findings in detail in discussion section.
  2. In Fig 2C., It is suggested to replace the better images for DAPI staining.
  3.  Authors should provide the animal images and excised tumors for in vivo study as it will provide the better understanding.

Round 2

Reviewer 1 Report

The authors have clarified several of the questions I raised in my previous review. Most of the major problems have been addressed by this revision.